# Applications of Long-Length Carbon Nano-Tube (L-CNT) as Conductive Materials in High Energy Density Pouch Type Lithium Ion Batteries

**DOI:** 10.3390/polym12071471

**Published:** 2020-06-30

**Authors:** Shan-Ho Tsai, Ying-Ru Chen, Yi-Lin Tsou, Tseng-Lung Chang, Hong-Zheng Lai, Chi-Young Lee

**Affiliations:** 1Department of Materials Science and Engineering, National Tsing Hua University, Hsinchu 300, Taiwan; azwupa23@gmail.com (S.-H.T.); annieblack05@gmail.com (Y.-R.C.); leelab212@gmail.com (Y.-L.T.); 2Foxconn Technology, Group Shih Hua Technology, LTD., Hsinchu 300, Taiwan; wimbly.tl.chang@cntouch.com (T.-L.C.); leo.hc.lai@cntouch.com (H.-Z.L.)

**Keywords:** high energy density LIB, pouch type LIB, long-length carbon nano tube (L-CNT), carbon black, ionic conductivity

## Abstract

Lots of lithium ion battery (LIB) products contain lithium metal oxide LiNi_5_Co_2_Mn_3_O_2_ (LNCM) as the positive electrode’s active material. The stable surface of this oxide results in high resistivity in the battery. For this reason, conductive carbon-based materials, including acetylene black and carbon black, become necessary components in electrodes. Recently, carbon nano-tube (CNT) has appeared as a popular choice for the conductive carbon in LIB. However, a large quantity of the conductive carbon, which cannot provide capacity as the active material, will decrease the energy density of batteries. The ultra-high cost of CNT, compared to conventional carbon black, is also a problem. In this work, we are going to introduce long-length carbon nano-tube s(L-CNT) into electrodes in order to design a reduced-amount conductive carbon electrode. The whole experiment will be done in a 1Ah commercial type pouch LIB. By decreasing conductive carbon as well as increasing the active material in the positive electrode, the energy density of the LNCM-based 1Ah pouch type LIB, with only 0.16% of L-CNT inside the LNCM positive electrode, could reach 224 Wh/kg and 549 Wh/L, in weight and volume energy density, respectively. Further, this high energy density LIB with L-CNT offers stable cyclability, which may constitute valuable progress in portable devices and electric vehicle (EV) applications.

## 1. Introduction

Rechargeable lithium-ion batteries (LIB) are widely applied as power sources in many consumer and portable electric devices due to their significant advantages, such as specific working voltage and high energy density. Recently, LIB has become commercially available as one of the promising choices for energy storage systems (ESS) and electric vehicles (EV) [1,2,3]. In the past few decades, the cathode (positive electrode) of rechargeable LIB has been largely based on the lithium metal oxide component, especially layer types LiCoO_2_ (LCO) and LiNi_x_Co_y_Mn_1-x-y_O2 (LNCM) [4,5]. Conventionally, LCO is used in mobile phones and laptop computers with the highest energy density requuirements, and LNCM is generally applied in EV because of the cost and safety issue. Nevertheless, in both LCO and LNCM, during electrochemical reactions, the oxidation and decomposition of electrolytes results in an interaction on the surface of electrodes, causing defects and capacity decay of the electrode and the cell. For this reason, surface coating is one of the most effective ways to improve the stability of the electrode surface in order to improve the cell cycle life. Most of the research shows that the surface coatings demonstrated to improve the performance of LIBs are metal oxides, such as Al_2_O_3_, SnO_2_, ZrO_2_, MgO and ZnO [6,7]. Metal oxides and metal fluorides are usually highly stable and electrically non-conductive, and therefore increase the internal resistance of the lithium cells [8,9]. In this regard, highly conductive carbon material, with stable properties and reasonable cost, has become known as an effective approach to improving the conductivity of the electrodes [10,11]. Carbon, including graphites such as KS4 and KS6 (TIMICAL), and carbon black, such as acetylene black (AB), ketjen black (KB) and SuperP-Li (TIMICAL), are comparatively good choices for mixing into LIB electrodes, with reasonable cost, excellent conductivity and stable electrochemical characteristics. Besides them, recently, carbon nanotubes (CNTs), because of their unique 1D tubular structure, and high electrical and thermal conductivities, have been used as additives in both anode and cathode materials, to satisfy such multi-functional improvements as (1) increased Li-ion intercalated/removal rates, due to the short transport path, and (2) enhanced electron transport properties for LIBs [12,13,14]. However, large amounts of conductive carbon and relevant electrochemical inactive binders contribute no capacity to a cell. The proportion of (lithium-provided) active material decreases as inactive parts increase, resulting in low energy density (Wh/kg or Wh/L) battery products. How to decrease the usage of additives while maintaining stable battery performance has become an important issue in the LIB industry.

The price of CNT as a conductive additive is over 30 times more expensive than carbon blacks. However, due to its proven effectivity, such as resistance reducing in electrodes and performance improvement in the cells [15], multi-wall carbon nano tubes (MWCNT) are considered as highly plausible commercialized products, and have been widely applied in high-niche LIBs. Companies like C-Nano and Dynanoic have high marketing shares in providing MWCNT as a conductive carbon to global LIB companies. The length of a commercial MWCNT product is around 1–3μm after dispersion into the electrode. Generally, the applied amount of MWCNT is 0.5–1%, taking the place of the traditional 2–2.5% carbon black, or combining with the carbon black in the LIB electrode to improve the battery performance.

In this research, a long-length MWCNT (L-CNT) is introduced as new conductive additive to LIB. The approximate length of the original L-CNT is 100–250 μm, and the dispersed L-CNT inside the electrode is 15–25 μm. We assume that the long-length L-CNT may make it easier to make a network inside the electrode, compared to the traditional MWCNT. After the high-quality L-CNT is dispersed into the LNCM positive electrode of the Li-ion cells, the electrochemical experiments can be done in a 1Ah pouch type LIB product by paring with the graphite negative electrode. Since high cost is one of the main shortcomings of CNT, we used only 0.16% of L-CNT as a conductive additive in the LNCM cathode, which is about 1/3 of the MWCNT presently needed.

The aim of this study is to examine the effect of a very small amount of L-CNT as a conductive additive in the LNCM pouch LIB. Electrochemical performances, including charge–discharge, C-rate and long cycle life testing of the pouch cell, will be studied. The resistivity and impedance test will also be done to support the results. By decreasing the amount of conductive carbon, the ultra-high active material (> 97.8%) contained in the electrode may generate high energy in the cell. The energy density (in Wh/kg, Wh/L) and capacity for retention are also attractive goals we are going to observe. With the small amount of effective L-CNT applied, the industry could control the reasonable cost of conductive additives and increase the business value of LIB products.

## 2. Experimental

### 2.1. Characteristics of Long-Length CNT (L-CNT)

The as-received L-CNT (Shih Hua Technology) sample was bulk-like solid as shown in Figure 1a. The crystalline structure of the L-CNT was studied by X-ray diffractometry (XRD, Philip PW-1700, Philips, Eindhoven, Netherlands) operated at 40 kV and 40 mA with Cu Kα radiation. The morphology and size of L-CNT were examined with a field emission scanning electron microscope (FESEM, JEOL JSM-6500F, JEOL, Akishima Tokyo, Japan). The structure and crystallization of graphite order of L-CNT was analyzed by Raman spectroscopy (HR800 UV, Horiba Jobin Yvon, Kyoto, Japan).

### 2.2. Analysis and Dispersion of L-CNT

After being dried at 120 °C for 4 h, 1 g of original L-CNT was added into 200 mL of N-Methyl-2-pyrrolidone (NMP) containing 2 g of polyvinylpyrrolidone (PVP) under 20 m/s dispersion, done by high speed mixer (FILMIX model-56, PRIMIX, Mijdrecht, Netherlands). The gel-like product was then pasted on a PET plate in a 3 cm × 3 cm area as shown in Figure 1b. After drying, film-like L-CNT samples were ready for analysis. The electrical resistance of the L-CNT sample was measured by the four-point probes equipment (2450 source meter, KEITHLEY, Cleveland, OH, USA).

### 2.3. Manufacture of Working Electrode

The positive electrode consisted of 97.84 wt % LiNi_5_Co_2_Mn_3_O_2_ (LNCM) (CITIC GUOAN, Beijing, China) as an active material, 2 wt % polyvinylidene difluoride (PVDF) (HSV-900, Arkema, Colombes, France) as the binder, and 0.16 wt % conductive carbon additive (L-CNT or carbon black super P-Li). L-CNT was as a paste made in Section 2.2, and the solid content was 0.5%. The negative electrode contained artificial graphite (China Steel, CSCC, Kaohsiung, Taiwan) (94 wt %), PVDF (4 wt %), and super P-Li (2 wt %). The slurry-preparing equipment was a triple-shaft planet disperser (HIVIS DISPER MIX model 3D-2). The NMP-based positive (cathode) and negative (anode) slurry were coated onto the Al foil and Cu foil by the comma coater (TM-MC, HIRANO TECSEED, Tokyo, Japan,) respectively. The electrodes were then vacuum-dried overnight at 110 °C. The approximate loadings of the cathode and anode were 3.25 and 3.68 mAh/cm^2^, respectively.

### 2.4. Electrochemical Property Tests of the Cells

A prismatic soft-packed pouch lithium-ion battery (1100 mAh), about 3.6–3.7 mm thick, 38 mm wide and 55 mm long (Model 404060), as shown in Figure 1c, was assembled using as-made LNCM cathodes, graphite anodes and polypropylene separator. It was a stacked type pouch cell containing 9 pairs of 36 mm × 52 mm cathode and anode electrode sheets constructed in zig-zag mode along with a separator as shown in Figure 1d. 1 M LiPF6 in EC/DEC/EMC (2:2:1 volume ratio) mixture solvent with 3% propylene carbonate (PC) and 1% vinylene carbonate (VC) was used as the electrolyte. The electrochemical tests were carried out using these laminated pouch LIBs assembled in a dry room with dew point temperature −40 °C. Pouch cells were sealed 30 min after electrolyte injection inside a dry room. Then it took 6 h for electrolyte soaking of the cells before the electrochemical tests. The electrochemical properties of the cells were tested with a multi-channel automatic battery cycler (Type 5V6A, NEWARE, Hong Kong, China) with a constant discharging current from 200 to 3000 mA. In the current rate performance tests, the cells were charged to 4.3 V with a constant current and constant voltage (CC/CV) at a current of 200 mA, and then discharged to 2.8 V. The cut-off current of CV charge was 1/10 of CC current. In the charge/discharge cycling test, the cell was charged and discharged at the same current (such as 500 or 1000 mA) repeatedly. The cut-off voltages were set at 4.3 and 2.8 V. OCV and ACIR (1000 Hz) measurements of the cells were done using an impedance analyzer (HIOKI-3555, HIOKI, Nagano, Japan). Variable frequency AC impedance measurements of the cells were conducted using an electrochemical measurement unit (Solartron Instruments, SI1280B), and the frequency range and voltage amplitude were set at 100 kHz to 0.01 Hz and 5 mV, respectively. The Code Z-View software was used to fit the spectra to the possible equivalent circuit.

## 3. Results and Discussion

### 3.1. Morphology and Characteristics of L-CNT

Figure 2a shows the morphology of the original L-CNT sample. Clustered L-CNT with a length of 100–200 μm are shown in the images. In Figure 2b, the estimated length of the flexible fiber-like L-CNT is over 110 μm. The thickness of the dried L-CNT film in Figure 1b is about 60–70 μm, and Figure 2c is the morphology of the dispersed L-CNT film-like sample. The length of the dispersed L-CNT is about 15–20 μm, and the diameter of the L-CNT is approximately 10nm, as shown in Figure 2d. The electrical resistivity of the L-CNT film is 0.0139 Ω cm, averaged from the values of nine pieces of film samples.

The XRD profiles of the L-CNT are illustrated in Figure 3a. It can be seen that the pattern of the L-CNT sample is almost the same as that of the crystalline graphite in (002) plane. The peak shows at 2θ = 25.85°, corresponding to an interplanar space of 3.44 Å, with a slight shift due to the curve of the graphite plane in the CNT. Raman spectroscopy is an appropriate method [16,17] for investigating the graphene layer in carbonaceous materials (including CNT). In Figure 3b, the left peak at 1352 cm^−1^, named D band, is induced by the defect and non-basal plane of the graphitic layer. In the middle part, first-order Raman spectrum of highly crystalline graphite shows a sharp band at 1591 cm^−1^, known as G band, associated with in-plane symmetric carbon Sp^2^ bonding. The peak has shifted to higher wavenumbers, compared to normal graphite at 1580 cm^−1^, indicating that there are less carbon plane sheet layers in the L-CNT than in the graphite. The broad peak at 2690 cm^−1^ is the second-order spectrum attributed to the overtone of D band, referring to a multi-wall CNT.

Figure 4 illustrates the morphology of the LNCM positive electrode containing different conductive carbon materials with the same wt % as that of the active material, conductive carbon (L-CNT or super P-Li), and the binder obtained after the slurry coating process. The formulation of the electrodes is 97.84% LNCM, 2% PVDF and 0.16% conductive carbon. The conductive carbon in the electrode [Figure 4a] is carbon black, named super P-Li. Super P-Li accumulates with PVDF particles in some places, making LNCM particles connect to others, as shown in the yellow circle of Figure 4a. However, most of the spaces between LNCM particles are empty, indicating that this amount of carbon black offers no help in making a connecting network between active materials. Furthermore, there is almost no super P-Li cover on the surface of LNCM. By contrast, in the electrode with 0.16% L-CNT as the conductive carbon, it is seen that L-CNT is dispersed between the particles and on the particle surface, to form a continuous network-like structure connecting and covering the LNCM particle, as shown by the yellow arrows in Figure 4b.

### 3.2. Electrochemical Properties of L-CNT-Containing LNCM Electrodes and Cells

After the characterization of the structure and chemical composition, L-CNT-containing, LNCM electrode-based 1Ah LNCM/graphite full pouch cells were fabricated to investigate the electrochemical properties. Figure 5a,b display the 200 mA charging (lithium intercalation into negative graphite) curve and discharged (lithium de-intercalation) curves under various current densities of the cell, with 0.16% carbon black super P-Li, and a 0.16% L-CNT-containing cathode. At the current of 200 mA, the discharged capacity of the cell with super P-Li is not able to reach 1Ah, which is 864.1 mAh. The cell containing L-CNT has a higher capacity; 1147.7 mAh. In Table 1, by dividing the weight of the active material in LNCM, the unit capacity performances of the cathode materials are 170 and 129 mAh^−1^g, in L-CNT and super P-Li cell, respectively, revealing an insufficient amount of carbon black super P-Li in the role of the conductive carbon. Observing the 200 mA charging and 200 mA discharging in Figure 5a,b, the discharging voltage plateau is higher and the charging voltage plateau is lower in the L-CNT-containing cell than in super P-Li-containing cell, indicating a lower internal resistance and higher ionic conductivity in the L-CNT-containing cell than in the cell containing the same amount of super P-Li. Table 1 compares the details of the electrode and pouch cells, including formulation, cell volume, weight and energy density. While every pouch cell share a similar volume and weight, the L-CNT-containing cells with a higher capacity reach a greater energy density (Wh/kg and Wh/L). The energy density of the 0.16% L-CNT-containing cell is 223.6 Wh/kg or 549.3 Wh/L, about 35% higher than the cell containing 0.16% super P-Li. The energy density would increase more in larger capacity pouch cells. To know more fully the practical utility of the cells containing different formulations, the higher discharging current of these LNCM/graphite pouch cells is also evaluated. In Figure 5a, the capacity of cells containing 0.16% super P-Li decreases rapidly with the current density, and is almost zero at the current of 3000 mA. In contrast, the discharging capacity of L-CNT-containing cathode at 500mA is 164.86 mAh/g, and 1113.3 mAh for the pouch cell, as shown in Figure 5b and Table 1. The capacities of the L-CNT-containing cell are 1063.3, 1013.5 and 848.5 mAh at 1000, 2000 and 3000 mA discharging current, respectively. The corresponding values for the super P-Li-containing cell are 297, 89 and 14 mAh, as summarized in Figure 5c.

The cell containing 0.16% L-CNT shows very high energy density, supported by a large capacity provided by the LNCM cathode, and a high voltage platform contributed by the effective conductive network built from L-CNT inside the electrode. The weight energy density of the L-CNT cell can still reach 205 Wh/kg under 1000 mA. That is, only 4.5–5 kg of cells could provide 1 kWh energy. The weight energy density and volume energy density of the 0.16% L-CNT-containing cell are both four times higher than those of the 0.16% super P-Li-containing cell under 1000 mA discharging, indicating increased ionic conductivity and decreased resistance in the electrode and cell, sustained by combining very high amounts of active material (LNCM) with a low amount of L-CNT.

Figure 6a shows the plot of discharge capacity as a function of the cycle number for the pouch cells containing different formulation of LNCM cathode. The current in this test is 1C, which is 864 mA and 1148 mA (defined by the capacity under 0.2 A) on the super P-Li cell and L-CNT cell, respectively, calculated from the capacity tested under 200 mA. It is obvious that the L-CNT-containing LNCM electrode cell exhibits a far better cycle life performance, compared with the super P-Li-containing electrode in this specific formulation. The capacity retention is 87% in the pouch cell with L-CNT after 500 cycles, whereas the super P-Li cell suffers serious capacity fading and shows only 40% capacity left within 50 cycles. During long cycling tests, the discharge of working mid-point voltage of the L-CNT cell keeps constant at 3.5–3.6 V, as shown in Figure 6b. By contrast, the super P-Li cell shows fast decay of working mid-point voltage, indicating a decreasing of ionic conductivity, possibly resulting in the sudden capacity fading. We believe that the bad ionic conductivity of the super P-Li electrode may due to (1) inadequate quantity of carbon black to make the connection between LNCM active materials at the first cycle, and (2) losing the particle-to-particle contact once made after cycling. The LNCM’s active material volume changes after long-term cycling, due to the low resilience of the sphere-like carbon black, which means the carbon black has no ability to maintain the contact between LCNM particles and other particles, thus forming a crack inside the electrode in microscopy. It was proved that this crack would change the microstructure of the electrode and generate electrical isolating between active materials, causing performance degradation including capacity and voltage fading [18,19].

Impedance of the cells was also recorded to show the differences between the two cathodes. The ACIR (1000 Hz) of the super P-Li-containing cell is 71.9 mΩ at OCV = 0.281 V, and that of the L-CNT-containing cell is 20.1 mΩ (0.286 V). It is evident that the difference in the resistance is caused by the composition of the LNCM cathodes, since all components in the cell are the same. The electrochemical impedance spectroscopy (EIS) test was applied to analyze the electrochemical behavior of LNCM/graphite pouch cells. Figure 7a exhibits the Nyquist plots of the 0.16% L-CNT- and 0.16% super P-Li-containing full cells at 50% state of charge (SOC) in the first cycle. In a typical EIS curve, the resistance measured at very high frequencies corresponds to the resistance of the ionic electrolyte ‘Re’, and is added in series to the equivalent circuit in Figure 7b. The high frequency and medium frequency, in a remarkably semicircle-like curve, refer to the resistance of Li^+^ migration through the SEI (solid electrolyte interface) layer and the charge-transfer resistance (Rct) on the particle surface. In general, the cell impedance increase is mainly caused by the charge-transfer resistance [20,21]. The capacitance of SEI film and the capacitance of the double layer on the particle surface are represented by constant phase elements (CPE) Qsei and Qdl, respectively [22,23,24]. By fitting the impedance Nyquist plot by Z-viewer software, the Re, Rsei and Rct values could be simulated. The equivalent series resistance, Rcell (value of Re + Rsei + Rct), could be referred as an index of the overall internal resistance of the cell [25,26]. The cell with the LNCM/PVDF/super P-Li = 97.84/2/0.16 cathode (and graphite anode) shows a little higher Re, but far higher Rsei and Rct, thus resulting in 4.75 times higher internal resistance than in the cell with the LNCM/PVDF/L-CNT = 97.84/2/0.16 cathode, as summarized in Table 2. This result implies that by adding same amount of 0.16% LCNT and SP as conductive carbon in the cathode, the LCNT forms a network between LNCM particles and disperses onto the LNCM surface to fabricate a continuous electron conductive pathway, for electrons to insert into and extract from the LNCM active material. Moreover, the low frequency region of the Nyquist plot displays an inclining line at about a 45 ° angle to the x-axis Z’ resistance. A Warburg impedance element (Wd) in Figure 7b is chosen to represent the bulk diffusion of lithium ions [27,28]. According to the semi-infinite diffusion model, the Warburg impedance Z’ could be expressed as kw ω^−1/2^ (1 − j) [24], involving the Warburg coefficient kw as a function of the slope of Z’ vs. ω^−1/2^ at the low frequency region, while ω is the angular frequency 2πf. The Z’ vs. ω^−1/2^ correlates to a ‘Randles plot’ for both 0.16% L-CNT and 0.16% super P-Li cathode cells, as shown in Figure 7c. The diffusion coefficient of Li-ion (D_Li_) for both kinds of cell could be calculated by the relation of kw, listed also in Table 2. The diffusion coefficient of the LNCM/PVDF/L-CNT = 97.84/2/0.16 cathode (graphite anode) pouch cell is two orders higher than that of the LNCM/PVDF/super P-Li = 97.84/2/0.16 cathode cell, which is 6.25 × 10^−7^ vs. 2.86 × 10^−9^, indicating an improvement in ionic conductivity of the cathode. The improvement may be attributed to a faster removal of lithium ions from the LNCM surface through high porosity channels in the electrode [29,30]. L-CNT in this result is proved to effectively decrease Li-ion diffusion resistance and enhance the ionic transportation in order to increase the lithiation/de-lithiation rates, thus generating the good ionic conductivity of the electrode and the cell with very minimal addition.

The AC impedance tests of the super P-Li and L-CNT cells were also measured at the end of the cycle life test, as shown in Figure 8. In Figure 8a, the LNCM/PVDF/superP-Li = 97.84/2/0.16 cathode (anode graphite) pouch cell increases the internal resistance ‘Rcell’ approximately threefold, from 0.57 Ω to 1.8 Ω, after only 37 cycles. In contrast, the Rcell of the LNCM/PVDF/L-CNT = 97.84/2/0.16 cathode only slightly increases with cycles, and reaches 0.163 Ω after 500 cycles at 1000 mA, as illustrated in Figure 8b, even lower than the initial Rcell of the LNCM/PVDF/superP-Li = 97.84/2/0.16 cathode cell. We believe that the L-CNT-containing LNCM cathode cell keeps a stable cycle life via good control of internal resistance, with an effective L-CNT conductive additive inside the electrode. In the case of the 0.16% super P-Li cathode, no conductive network was formed, giving no chance to improve the electric conductivity between active materials, resulting in poor C-rate and cycle performances.

The pouch cells were disassembled for cathode observation after the impedance test at the last cycle. Figure 8 also provides the SEM images of the cathode following disassembly from the pouch cells after the cycling test. Figure 8c shows the electrode containing 0.16% super P-Li after 37 charge–discharge cycles; the LNCM active materials separate from each other leaving an empty gap, indicating evidence of a high polarization response with a large resistance increase in the EIS test. On the other hand, it could be confirmed that the dispersion of the conductive network between the cathode active material in the LNCM is still obvious after 500 testing cycles. This 0.16% L-CNT-containing cathode exhibits a membrane-like structure covering the LNCM, as shown in Figure 8d. It provides small internal resistance change and great tolerance during cycling. The existence of the network-like and membrane-like construction clearly contributes to the activation of the cathodes by providing an effective conducting pathway for lithium ions.

In summary, L-CNT with enough length could build an effective conducting network inside LIB electrodes, to decrease the internal resistance and improve the ionic diffusion and transportation for the electrochemical reaction of the cell. Good electric conductivity, resulting from lower internal resistance and better ionic conductivity caused by a high diffusion coefficient, could increase the cycling rate and power density by alleviating the polarization and ineffectiveness of the electrode [31,32], thus significantly improving the rate capability and cyclic performance of the cells. Further, L-CNT shows a unique morphology and high mechanical strength, and could decrease the chance of pulverization of the conductive network during long-term electrochemical reaction.

## 4. Conclusions

In this study, long-length (> 100 μm, originally) carbon nano-tube (L-CNT) has been successfully dispersed into a LiNi_5_Co_2_Mn_3_O_2_ (LNCM) cathode to make a LNCM-Graphite pouch cell. The L-CNT forms a network-liked frame structure between the active materials at the beginning, which exists until the end of the charge–discharge cycling tests. The cells with the cathode containing 97.84% LNCM and 0.16% L-CNT indicate good C-rate tolerance and cycle ability, compared to the cell containing the same amount of carbon black (super P-Li). The enhanced performance is attributed to L-CNT, which stabilizes the structure of the electrode, including the interaction between the active materials, in order to guarantee good electric conductivity and ionic conductive pathways, in order to sustain the long cycle ability under a specific cycling rate. To clarify, the 500 repetitions of the 1C (100% discharged in 1 h) cycling test condition in this study meets the industry standard, indicating a valuable design for commercial LIBs. Moreover, with small and effective amounts of L-CNT as the conductive additive, the cell could reach higher energy density (Wh/kg and Wh/L) accompanied with reasonable costs in the additive and production processes.

## Figures and Tables

**Figure 1 polymers-12-01471-f001:**
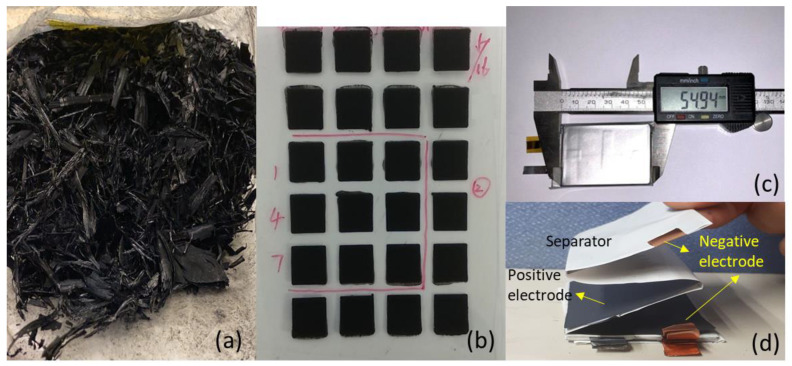
Photos of (**a**) original L-CNT as received, (**b**) L-CNT film on PET plate made by dispersed paste, (**c**) prismatic soft-packed pouch lithium-ion battery (~1100 mAh), and (**d**) internal components of a disassembled pouch cell.

**Figure 2 polymers-12-01471-f002:**
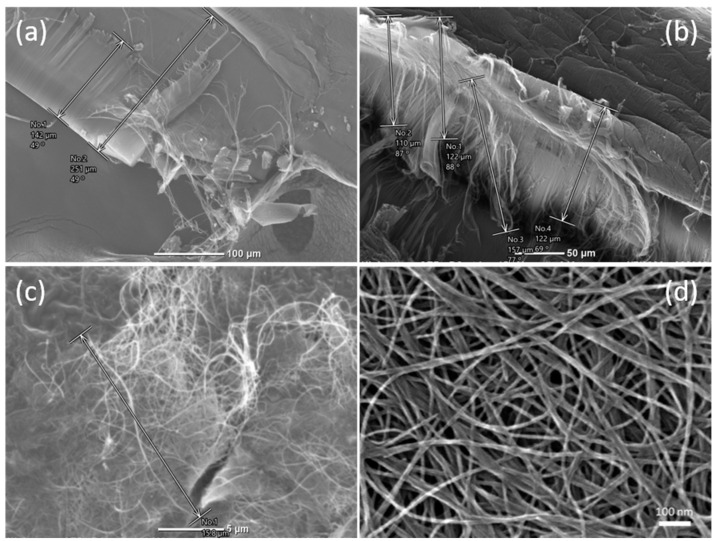
SEM images of (**a**,**b**) original L-CNT products, (**c**) L-CNT film coated by dispersed paste and (**d**) L-CNT film in higher magnification.

**Figure 3 polymers-12-01471-f003:**
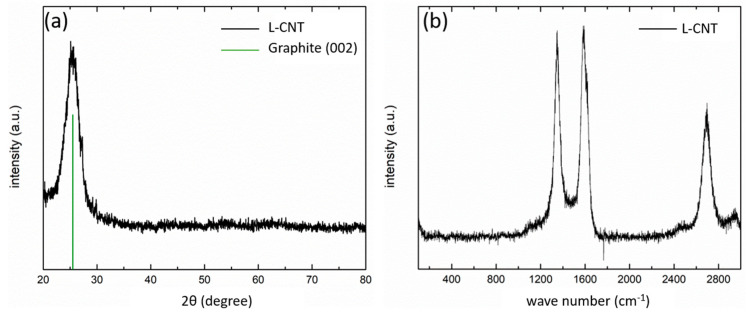
(**a**) XRD analysis and (**b**) Raman spectroscopy of original L-CNT.

**Figure 4 polymers-12-01471-f004:**
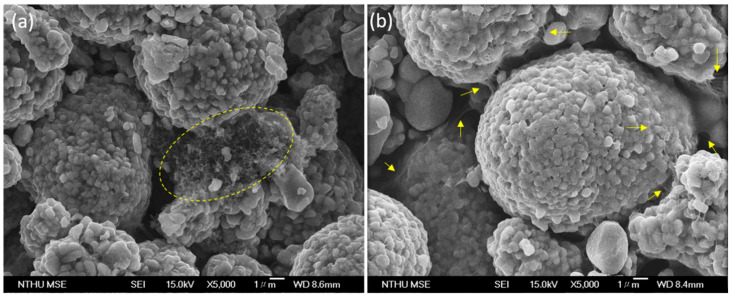
SEM of the positive electrodes (cathodes) with 97.84% of LNCM, 2% of PVDF and (**a**) 0.16% of super P-Li, (**b**) 0.16% of L-CNT.

**Figure 5 polymers-12-01471-f005:**
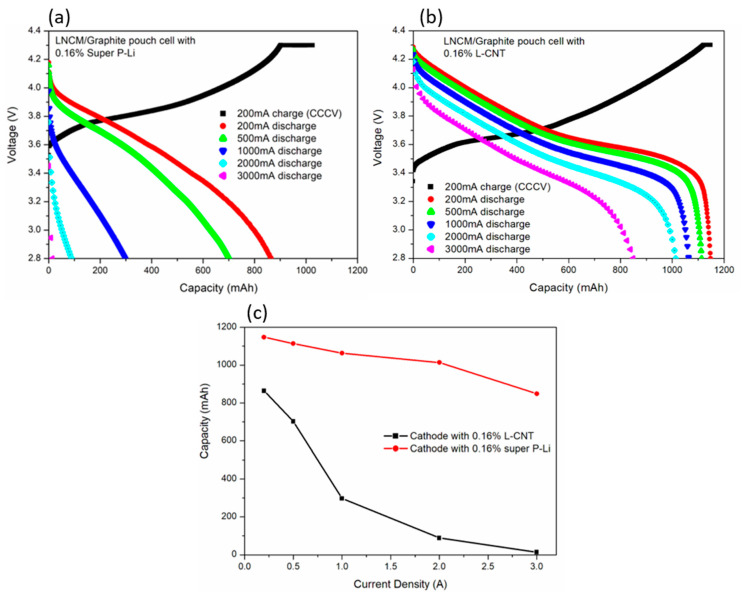
Current rate performance of LNCM/Graphite pouch LIB with (**a**) 0.16% super P-Li and (**b**) 0.16% L-CNT in the cathode. (**c**) Discharge capacity of the Super P-Li-containing cathode and L-CNT-containing cathode at various current rates.

**Figure 6 polymers-12-01471-f006:**
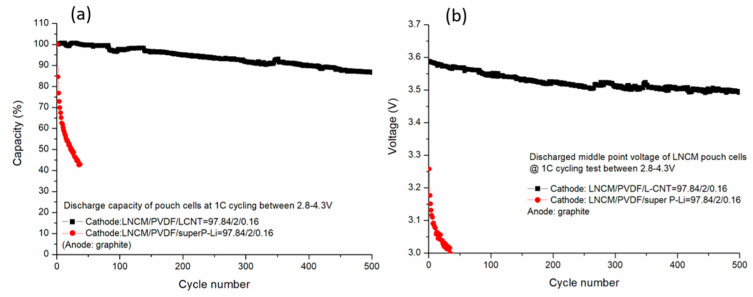
(**a**) 1C cycle life testing of the L-CNT-containing and the super P-Li-containing LNCM pouch cells. (**b**) Discharge of middle point voltage change during cycling test.

**Figure 7 polymers-12-01471-f007:**
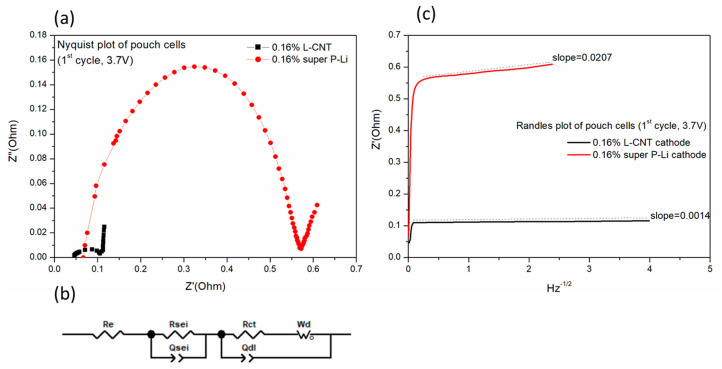
(**a**) Nyquist plots of LNCM/graphite pouch cell with LNCM/PVDF/L-CNT = 98.84/2/0.16 and LNCM/PVDF/super P-Li = 98.84/2/0.16 as cathode, (**b**) the equivalent circuit for describing AC impedance of the cells between 100kHz and 0.01Hz, and (**c**) the Randle plots of the cells at 50% SOC, first cycle.

**Figure 8 polymers-12-01471-f008:**
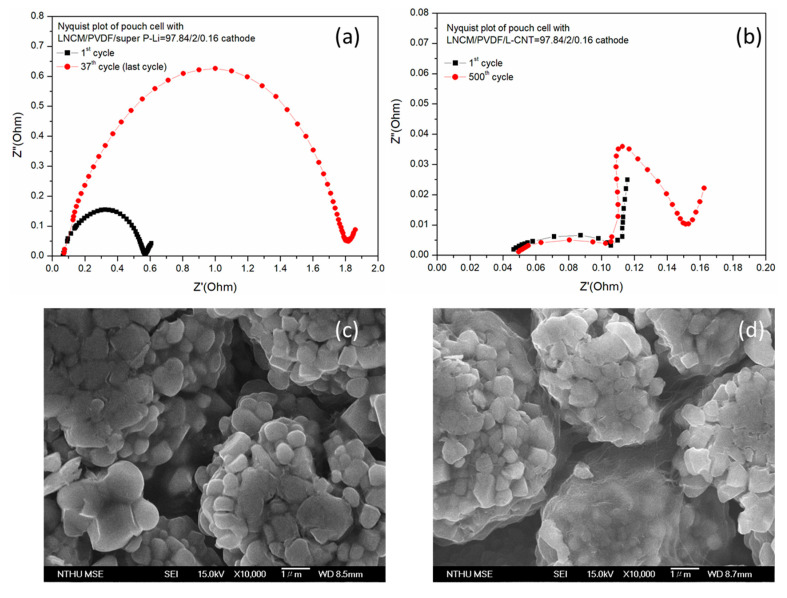
Nyquist plots of the (**a**) 0.16% carbon black super P-Li-containing cathode cell and (**b**) 0.16% L-CNT-containing cathode cell at the first and the last cycle. SEM images of the cathodes (**c**) LNCM/PVDF/Super P-Li = 97.84/2/0.16 after 1C, 37 cycles and (**d**) LNCM/PVDF/L-CNT = 97.84/2/0.16 after 1C, 500 cycles in LNCM/Graphite pouch LIBs.

**Table 1 polymers-12-01471-t001:** Effect of discharge current on the capacity and energy density of the lithium-ion pouch cells containing different positive (LNCM) electrodes.

Cell No.	200 mA Discharge		500 mA Discharge		1000 mA Discharge		2000 mA Discharge		3000 mA Discharge	
	Capacity (mAh, mAh/g)	Energy density (Wh/kg) (Wh/L)	Capacity (mAh, mAh/g)	Energy density (Wh/kg) (Wh/L)	Capacity (mAh, mAh/g)	Energy density (Wh/kg) (Wh/L)	Capacity (mAh, mAh/g)	Energy density (Wh/kg) (Wh/L)	Capacity (mAh, mAh/g)	Energy density (Wh/kg) (Wh/L)
SP 016	864.1, 129.4	164.2 405.7	702.5, 105.2	132.0 326.2	297.2, 44.50	51.51 127.2	88.8, 13.30	14.41 35.59	13.9, 2.081	2.19 5.41
LCNT 016	1147.7, 170.0	223.6 549.3	1113.3, 164.96	218.5 536.9	1063.3, 157.46	205.3 504.5	1013.5, 150.08	191.2 469.1	848.5, 125.65	157.0 385.7

**Table 2 polymers-12-01471-t002:** Impedance parameters of pouch cells with the L-CNT- and the super P-Li-containing cathodes in the first cycle.

Cell (Electrode)	Re (Ω)	Rsei (Ω)	Rct (Ω)	Rcell (Ω)	D_Li_ (cm^2^s^−1^)
LCNT-LNCM/Graphite	0.035	0.015	0.061	0.11	6.25 × 10^−7^
Super P-LNCM/Graphite	0.067	0.083	0.42	0.57	2.86 × 10^−9^

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
