# Peer review of "Applications of Long-Length Carbon Nano-Tube (L-CNT) as Conductive Materials in High Energy Density Pouch Type Lithium Ion Batteries"

_polymers, 2020, doi:10.3390/polym12071471_

Round 1

Reviewer 1 Report

The authors appear to demonstrate that composite electrodes containing 0.16% of carbon nanotubes perform much better at high rate discharge and cycle life than when carbon black is used.

-This difference is attributed to CNT's "uniformly dispersed on the particle surface" p5L156, which does not appear true of SuperP. Why would a nonpolar component be expected to uniformly disperse on a polar surface? While I appreciate the authors attempt to label the CNTs in Figure 4B, however, I am skeptical that they have succeeded in finding 0.16% of the electrode by mass when there also exists an order of magnitude more (2%) PVDF, which is going to be fibrous as well. This possibility could be the ultimate reason for the different electrochemistry observed: e.g., not so much the CNT/LNCM interaction but rather the CNT/PVDF interaction.

-Impedance: Data for cells before any electrochemical test should be shown; the earliest provided is during 50% state of charge. If the electrode is more conductive one should see this before any charging/discharging occurs. Also please provide some literature citations to compare the experimental DLi.

-Experimental: Please indicate the amount of time cells were aged, if any, between preparation and cell testing. If there really is even C dispersion on the surface of these particles in one case vs. another, one would suspect different kinetics of SEI formation and/or different equilibration times. If differences were substantial this could be monitored by OCV vs. time plots. If the SEI has formed to a different degree on one electrode vs. the other in correlation with subsequent electrochemical properties such an effect should be reported.

-There are several instances where the composites add to 101%, e.g. p8 L 242,244.

Reviewer 2 Report

Tsai et Al. present extensively and very well the using of L-CNT as component in comparison with carbon black named SuperP-Li in the cathodes of pouch LIB.

I really appreciate the design of the figures and the shape of the article: the reasearch sounds clear and neat, I really enjoy to read your article.

My suggestion is to pubblish after correcting few minor things and typos that I will list here below:

line 16: avoid the contractions, expecially in the abstract: cannot or can not are better

line 24: declare the acronyms, EV (even if known) for some reader result difficult

line42: formulas have not subscript items in line 34 they have, better be homogeneous

line113: as a non expert in the LIB I'd like to have a reference or a scheme or a figure explaining the zig-zag

line117: check the temperature spelling

line151_figure4: the yellow items you pointed out in the figure are not cited clear in the text neither in the figure caption, as you cleverly help the reader please write them down

thank you for your time

Reviewer 3 Report

Attached here is my review of the manuscript.

Round 2

Reviewer 1 Report

comments were addressed